# The Use of Gemtuzumab Ozogamicin as Salvage Therapy in Patients with Acute Myeloid Leukemia: A Monocentric Real-World Experience

**DOI:** 10.3390/medicina59030498

**Published:** 2023-03-02

**Authors:** İbrahim Halil Açar, Birol Guvenc

**Affiliations:** Division of Hematology, Department of Internal Medicine, Cukurova University, 01380 Adana, Turkey

**Keywords:** acute myeloid leukemia, gemtuzumab ozogamicin, targeted therapy

## Abstract

*Background and Objectives*: Relapsed or refractory acute myeloid leukemia (r/r AML) is a disease with a poor prognosis. Limited treatment options are available in r/r AML. Here, we administered gemtuzumab ozogamicin (GO) as salvage therapy in twenty-four patients with r/r AML. The aim of the study was to determine the role of GO in r/r AML in real life. *Material and Methods*: This retrospective observational study recruited 24 adult patients with diagnosed r/r AML from 2018 to 2022. Twenty-four patients with r/r AML were treated with GO. GO treatment was used as monotherapy in 23 patients and in combination with cytarabine in 1 patient. At the time of diagnosis, the risk status of all patients was determined as favorable, intermediate, or adverse according to the 2017 ELN AML guidelines. *Results*: The median follow-up was 44.3 (13–144) months. Fifteen (62.5%) of the twenty-four patients were in the intermediate-risk cytogenetics group and nine (37.5%) were in the favorable cytogenetics group. The most common adverse events included nausea/vomiting in 79.17% (*n* = 19) of patients, headache in 62.50% (*n* = 15), elevated LFTs in 37.50% (*n* = 9), febrile neutropenia in 25% (*n* = 6), and bleeding in 25% (*n* = 6). The most common cause of death was infection. The most common causes of mortality were septic shock, accounting for 33.3% (*n* = 8) of deaths, and opportunistic lung infection, accounting for 12.5% (*n* = 3) of deaths. Acute infusion-related toxicities associated with GO were usually transient and, in most cases, responded to the standard of care treatment. After treatment with GO, 16.6% (*n* = 4) of patients achieved MLFS and 37.5% (*n* = 9) achieved CR. The overall response rate was 54.1%. The median overall survival time of the patients was 44 months (37.8–50.2 months). Disease-free survival was 22 months (0–48.6 months). The 5-year survival rate was 33%. *Conclusions*: A low dose of GO improved the overall survival and disease-free survival in r/r AML patients. GO treatment had a positive safety profile in terms of toxicity.

## 1. Introduction

Acute myeloid leukemia (AML), which is characterized by the clonal expansion and differentiation arrest of myeloid progenitor cells, is the most common form of acute leukemia in adults [1,2]. It is generally prevalent among older adults, with a median age at diagnosis of 68 years [2]. AML accounts for the highest percentage of leukemic deaths among all types of leukemia; the 5-year survival rate is 28.7% [3]. Older age is associated with shorter survival compared with younger patients [4].

The main goals in the treatment of AML are to keep the disease under control and, if possible, to eliminate the disease. The European LeukemiaNet (ELN) 2017 report has led to significant advances in the diagnosis and management of AML treatment. Since the ELN 2017 report, cytogenetics, molecular markers, and minimal residual disease (MRD) have become very important in both prognosis and treatment planning [5]. Standard care is determined by testing new treatments in clinical trials and making comparisons with standard treatment. For fit individuals, induction therapy with cytarabine and anthracycline has been standard care for AML for many years. Over the years, several clinical trials have explored new treatment approaches, one of which is gemtuzumab ozogamicin (GO), to improve clinical outcomes. GO is a humanized anti-CD33 monoclonal antibody covalently linked to a semisynthetic derivative of calicheamicin, a potent cytotoxic antibiotic [6]. The demonstration of complete remission (CR) or complete remission with incomplete recovery (CRi) in approximately one-third of adult patients in early-phase clinical trials led the US Food and Drug Administration (FDA) to grant accelerated approval for the use of GO in the treatment of AML in 2000 [7]. However, it was withdrawn from the market because a post-approval phase III trial demonstrated no benefit of GO combined with chemotherapy over chemotherapy alone, as well as an increase in treatment-related mortality [8]. Subsequent studies have focused on several different dosing and administration strategies for GO to improve its safety profile while maintaining its clinical efficacy [9]. Several studies have shown that lower doses of GO in a fractionated dosing schedule were safer and more beneficial in patients with AML. In 2017, the FDA re-approved the use of GO for the treatment of newly diagnosed AML and r/r AML in adults [10,11]. In the ELN 2022 guideline, GO is recommended in patients with CD33-positive AML at favorable or intermediate cytogenetic risk [12].

The recurrence of AML after complete remission (CR), despite intensive therapy, is a major obstacle in approximately 60% of younger patients and 80–90% of elderly patients [13]. There is no specific salvage regimen as the standard of care treatment in patients with r/r AML. It has been shown by both phase I/II clinical trials and retrospective studies that GO alone or in combination with various regimens was safe and effective in patients with the first relapse of AML [14,15,16,17,18,19,20,21]. There are limited real-world data about the effect of GO on clinical outcomes in patients with recurrent AML [22,23]. Moreover, the use of GO in multiple relapses of AML patients has not yet been extensively investigated.

The aim of this study is to explore the clinical outcomes in patients with r/r AML who received GO as salvage therapy and to assess the mean disease-free survival rate.

## 2. Materials and Methods

This was a retrospective, single-center, observational study. Data were analyzed from 24 patients with r/r AML, who were older than 18 years of age, treated with GO as salvage therapy for multiple r/r, and diagnosed according to the WHO 2016 Acute Leukemia Diagnostic Criteria in the Hematology Service of the Department of Internal Medicine, Faculty of Medicine, Çukurova University between 2018 and 2022. Morphology, immunophenotyping by flow cytometry, conventional cytogenetic analysis, and gene mutation studies such as *FLT3-ITD*, which may be associated with AML, were performed from bone marrow aspiration samples. At the time of diagnosis, the risk status of all patients was determined as favorable, intermediate, or adverse according to the 2017 ELN AML guidelines. Those with a CD33 positivity above 20% in immunophenotyping by flow cytometry were considered CD33(+). Patients under the age of 18 years who were diagnosed with APL and those who did not receive GO were excluded.

In this retrospective study, GO treatment was administered as monotherapy in 23 patients and in combination with cytarabine (HiDAC; cytarabine 3 g/m^2^ over 3 h for days 1, 3, and 5) in one patient. All patients were administered GO treatment at a dose of 3 mg/m^2^/day on days 1, 4, and 7.

The treatment response rates were determined according to the ELN 2017 AML guidelines. The side effects were determined according to the FDA-approved drug safety information form for GO [24].

The aim of this study was to describe the efficacy and tolerability of GO therapy in patients with r/r AML.

### Statistical Analysis

The statistical analysis was performed using Statistical Package for the Social Sciences (SPSS), Windows v22.0 (IBM Corp., Armonk, NY, USA). The continuous variables were summarized using the mean and standard deviation in the event of normal distribution and using the median and interquartile range in the event of a non-normal distribution. The Kolmogorov–Smirnov test was used to assess the normality assumption. The continuous variables were analyzed using an independent *t*-test (and Welch *t*-test) or the Mann–Whitney test (under non-normality); *p* < 0.05 was taken as statistically significant. The Kaplan–Meier survival curves were plotted to compare the intergroup survival rates. The data were expressed as the mean ± SD (standard deviation) and the numbers as %. The McNemar test was used to compare the survey data.

## 3. Results

This retrospective study included a total of 24 patients with AML, older than 18 years of age, who were diagnosed and classified according to the WHO 2008 diagnostic criteria and treated with GO as salvage therapy for multiple r/r AML between 2018 and 2022. The mean follow-up was 44.3 (13–144) months.

Of these 24 patients, 8 (33.3%) were female and 16 (66.7%) were male. The mean age of the patients at diagnosis was 47.63 ± 16.93 (18–74) years; the mean age was 50.75 ± 15.68 years in females and 46.06 ± 18.33 years in males. There was no statistically significant difference in the mean ages at diagnosis between male and female patients (*p* = 0.57). The demographic characteristics of the patients are shown in Table 1.

The number of patients with pre-transplant relapsed AML and primary refractory AML was 22 (91.6%) and 2 (8.4%), respectively.

The dose of GO was 3 mg/m^2^, which was administered on days 1, 4, and 7. GO was used in combination with cytarabine (HiDAC; cytarabine 3 g/m^2^ over 3 h for days 1, 3, and 5) in one patient. Other concomitant medicines used were pheniramine hydrogen maleate, dexamethasone, and paracetamol.

The history of prior infections included herpes infections in 29.17% (*n* = 7) of patients, cytomegalovirus in 16.67% (*n* = 4), and pneumonia in 8.33% (*n* = 4). The comorbid diseases included diabetes, hypertension, chronic renal failure, and chronic obstructive pulmonary disease (COPD) in 29.17% (*n* = 7), 25% (*n* = 6), 8.33% (*n* = 2) and 8.33% (*n* = 2) of patients, respectively. Microscopic hematuria and proteinuria were common in 41.67% (*n* = 10) of patients in the urinalysis samples before treatment with GO. The PA chest X-ray showed bilateral infiltration (pneumonia) and opportunistic lung infection in 37.5% (*n* = 9) of patients, and the ECG demonstrated inferior MI and atrial fibrillation in 8.33% (*n* = 2) of patients.

The physical examination findings and *p* significance values before and after treatment with GO are provided in Table 2. Pallor was significantly less common in patients treated with GO (*p* = 0.008).

The biochemical outcomes of patients before and after GO treatment are shown in Table 3. While there were increases in the hemoglobin and AST levels after GO treatment, statistically significant decreases were observed in the peripheral smear and bone marrow blast rate.

The median overall survival time of the patients was 44 months (37.8–50.2 months). The 5-year survival rate was 33%. The disease-free survival was 22 months (0–48.6 months). The two-year disease-free survival rate was 45% (Figure 1).

The mean time from diagnosis to relapse was 21 months. After treatment with GO, 16.67% (*n* = 4) of patients achieved a morphologic leukemia-free state (MLFS), 37.5% (*n* = 9) achieved CR (complete remission), while 45.83% (*n* = 11) had progressive disease (PD). The overall response rate was 54.1%. Two of the patients who achieved a response later relapsed. Twelve (50%) patients underwent HSCT. The most common causes of death were infections. The causes of mortality are shown in Table 4.

The cytogenetic analysis showed that 24 patients had a normal karyotype (46 XX or 46 XY). Overall, fifteen (62.5%) were in the intermediate-risk cytogenetic group and nine (37.5%) were in the favorable cytogenetic group. Six (25%) female patients had intermediate risk while two (8.3%) had favorable risk; nine (37.5%) male patients had intermediate risk and seven (29.2%) had favorable risk. In our patient population, there were no cases with poor prognostic factors such as *FLT3-ITD*. The most common molecular mutations in the cases were *DNMT3A* 3 (12.5%), *biallelic CEBPA* 2 (8.3%), and *NPM1* 2 (8.3%). The immune phenotyping and cytogenetics are shown in Table 5.

The adverse events that occurred during the study period are shown in Table 6. Acute infusion-related toxicities associated with GO were usually transient and, in most cases, responded to standard of care treatment.

## 4. Discussion

AML is the most common form of aggressive leukemia and multiple treatment options are available to treat the disease. Although there are many intensive salvage treatment options for the treatment of AML, high mortality remains a major problem in patients with r/r AML who have an increased risk of toxicity due to multiple exposures to intensive chemotherapy. Low-dose GO has an acceptable toxicity profile in patients with r/r AML, which has resulted in its increased use, particularly in frail patients who have already received intensive therapy [25]. Acute infusion-related toxicities associated with GO are usually transient and respond to standard of care treatment [26].

GO is an immuno-conjugate that combines an anti-CD33 monoclonal antibody with a highly cytotoxic antibiotic, calicheamicin. It was first developed as a single agent for adults with recurrent AML, and then evaluated in combination with chemotherapy in newly diagnosed patients [27].

A recent meta-analysis confirmed that adding GO to chemotherapy may offer a survival benefit in patients who did not have adverse-risk cytogenetics [28]. However, these results were obtained regardless of the level of blast CD33 expression. Nevertheless, in vitro, a clear association was shown between CD33 expression and GO activity [29,30]. Conflicting in vivo results have been reported so far. There was no effect found when CD33 expression was considered as a continuous variable or by using a 20% cutoff.

After treatment with GO, the mean values of hemoglobin (g/dL), hematocrit (%), erythrocytes (M/µL), and AST (IU/L) were statistically significantly higher (*p* < 0.05) (Table 3). A comparison of pre- and post-treatment physical examination results showed a reduction in pallor and fatigue after treatment with GO. Although not statistically significant, there were slight increases in bleeding, respiratory distress, and renal failure. Hepatomegaly and splenomegaly costal arc length increased after treatment with GO, but no lymphadenomegaly was observed. Although the number of patients with pain remained the same, more patients treated with GO had widespread pain.

CD45, CD34, or CD117 are used to determine the blast rate in immunophenotypic studies [31]. The presence of CD56 antigen in blast cells affects the complete remission process and survival. The presence of CD56 in blasts in APL is considered a poor prognostic factor [28]. Immunophenotypic markers that have led to unfavorable and poor prognostic results in various studies include CD7, CD19, CD11b, CD13, CD14, CD33, CD34, CD56, and Tdt. The co-existence of CD34 and HLA-DR is an independent predictor of failure to achieve complete remission [29]. Tong et al. reported that in clinical trials, 96.4% (185/192) of the patients were CD13(+), 91.7% were (176/192) CD33(+), 83.9% were (161/192) MPO(+), 65.1% were CD34(+), 77.6% were HLA-DR(+), 26% were CD56(+), 20.8% were CD7(+), 9.9% were CD19(+), and 7.3% were CD2(+) (31). In this study, the immunophenotyping was as follows: CD33(+) in 100% (24/24), CD117(+) in 79.17% (19/24), MPO(+) in 70.8% (17/24), CD13(+) in 66.6% (16/24), and CD34(+) in 45.8% (11/24).

The median overall survival time in patients after treatment with GO was 44 months (37.8–50.2 months). The 5-year survival rate was 33%. The disease-free survival time was 22 months (0–48.6 months). The 2-year disease-free survival rate was 45% (Figure 1). In another study, the 5-year survival rate of AML was 28.7%, similar to the results in this study [3]. In a study conducted with 280 people, the median survival time after treatment with GO in 139 patients was 34 months (40.8%). The disease-free survival time was 28 months (50.3%) [32]. In one study, overall survival was not significantly affected by GO treatment in subgroups with a high CD33 [30], whereas in another study, the benefit associated with GO treatment in patients with high CD33 was still evident even after adjustment for cytogenetics and genotype [33].

According to the WHO classification, a blast rate above 20% is required for the diagnosis of AML (WHO). When peripheral smear and bone marrow blast percentages were considered, a statistically significant reduction occurred after treatment with GO. The median (min.–max.) value of the peripheral smear blast (%) was 55.00 (8–100) before GO treatment versus 0.5 (0.0–90.0) after treatment with GO (*p* = 0.000). The median (min.–max.) bone marrow—blast (%) level was 80.0 (20–100.0) before GO treatment versus 8.0 (1.0–95.0) after treatment with GO (*p* = 0.000).

In this study, the median platelet count (µ/uL) was 47,500 (9000–275,000) before treatment with GO versus 46,000 (3000–287,000) after treatment with GO. As for thrombocytopenia, there was no statistically significant difference between the pre- and post-treatment levels (*p* = 0.715).

Considering the cytogenetic risk assessment of the patients, 62.5% (*n* = 15) were in the intermediate-risk group while 37.5% (*n* = 9) were in the favorable-risk group.

The most common adverse events included nausea/vomiting in 79.17% (*n* = 19) of patients, headache in 62.50% (*n* = 15), elevated LFT in 37.50% (*n* = 9), febrile neutropenia in 25% (*n* = 6), and bleeding in 25% (*n* = 6). The most common causes of mortality were septic shock in 33.3% (*n* = 8) of patients and opportunistic lung infection in 12.5% (*n* = 3).

The administration of GO in patients who received multiple intensive treatment before or in combination with other agents at high doses resulted in increases in the toxic effects and the incidence of hepatic veno-occlusive disease (VOD) [34]. In this study, all patients received GO monotherapy (*n* = 23, 95.8%) or combination treatment (*n* = 1, 4.2%). As most patients had a history of multiple courses of intensive chemotherapy, GO was administered at a low dose (3 mg/m^2^) and intermittently (on days 1, 4, and 7) to avoid possible toxicities and to avoid increased risk of VOD.

## 5. Conclusions

In conclusion, a low dose of GO improved the overall survival and disease-free survival in r/r AML patients. In this study, GO was used as salvage therapy, and it appeared to have a positive safety profile in terms of toxicity. Considering that GO was a bridge therapy for allo-HSCT in half of the patients, it seems highly impressive that it offered a median survival of 44 months for patients with r/r AML. We believe that studies on the use of GO in the first-line treatments will further prolong the overall survival and the disease-free survival. In addition, it may be beneficial to conduct prospective studies with more patients in the future.

## Figures and Tables

**Figure 1 medicina-59-00498-f001:**
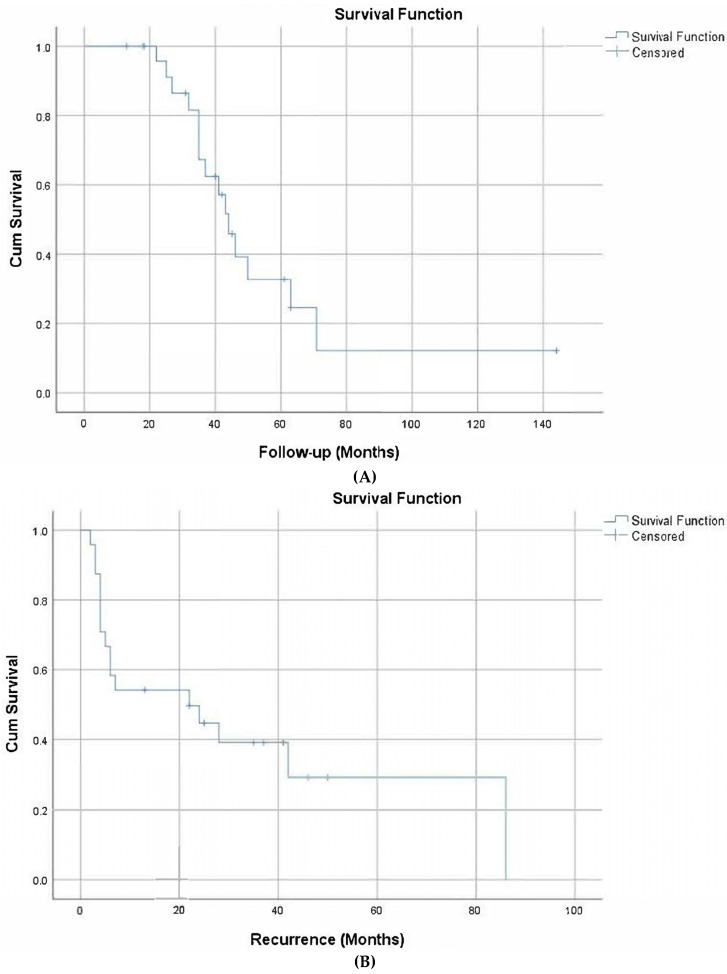
Overall survival (**A**) and disease-free survival (**B**).

**Table 1 medicina-59-00498-t001:** The demographic characteristics of patients.

	Overall, Mean ± SD	Female, Mean ± SD	Male, Mean ± SD
Age, years	47.63 ± 16.93	50.75 ± 15.68	46.06 ± 18.33
Marital status			
Married	21 (87.5%)		
Single	3 (12.5%)		
Education			
Illiterate	3 (16.7%)	3 (37.5%)	0 (0%)
Literate	1 (4.17%)	0 (0%)	1 (6.25%)
Elementary	7 (29.17%)	2 (25%)	5 (31.25%)
Secondary	5 (20.83%)	3 (37.5%)	2 (%12.5%)
High school	7 (29.17%)	0 (0%)	7 (43.75%)
University	1 (4.17%)	0 (0%)	1 (6.25%)

**Table 2 medicina-59-00498-t002:** Physical examination findings of the patients before and after treatment with GO.

	Before GO Treatment		After GO Treatment		
	Yes	No	Yes	No	*p* Value
Pallor	21 (87.5%)	3 (12.5%)	12 (50%)	12 (50%)	0.008
Fatigue	24 (100.0%)	0 (0.0%)	14 (58.3%)	10 (41.7%)	1.000
Fever	15 (62.5%)	9 (37.5%)	15 (62.5%)	9 (37.5%)	1.000
Weight loss	11 (45.8%)	13 (54.2%)	14 (58.3%)	10 (41.7%)	0.453
Bleeding	8 (33.3%)	16 (66.7%)	10 (41.7%)	14 (58.3%)	0.774
Skin manifestations	10 (41.7%)	14 (58.3%)	10 (41.7%)	14 (58.3%)	1.000
Respiratory distress	8 (333%)	16 (66.7%)	12 (50%)	12 (50%)	0.63
Hepatomegaly	2 (8.3%)	22 (91.7%)	2 (8.3%)	22 (91.7%)	1.000
Splenomegaly	6 (25.0%)	18 (750%)	5 (20.8%)	19 (79.2%)	1.000
Lymphadenomegaly	4 (16.7%)	20 (83.3%)	0 (0.0%)	24 (100.0%)	1.000
Visual impairment	3 (12.5%)	21 (87.5%)	-	-	
Convulsion	2 (8.3%)	22 (91.7%)	-	-	
Plegia	1 (4.2%)	23 (95.8%)	-	-	
Renal impairment	6 (25.0%)	18 (75.0%)	5 (20.8%)	19 (79.2%)	1.000
Pain	12 (50.0%)	12 (50.0%)	12 (50.0%)	12 (50.0%)	1.000

**Table 3 medicina-59-00498-t003:** Significance values for biochemical parameters before and after treatment with GO.

	Before GO Treatment	After GO Treatment	*p* Value
Hemoglobin (g/dL)	8.17 ± 1.03	10.53 ± 2.65	0.000 *
Hematocrit (%)	23.9 ± 3.5	30.9 ± 8.3	0.000 *
Erythrocytes (M/µL)	2.74 (1.87–3.57)	3.41 (2.08–3.87)	0.001 *
Leukocytes (µ/uL)	2400 (300–170,000)	4400 (200–71,800)	0.819
Thrombocytes (µ/uL)	47,500 (9000–275,000)	46,000 (3000–287,000)	0.715
ALT (IU/L)	32 (10–148)	50 (15–269)	0.055
AST (IU/L)	27 (10–80,0)	38.5 (14–130)	0.038 *
Creatinine (mg/dL)	0.85(6–49.0)	0.855 (0.1–5.87)	0.761
CRP (mg/L)	16 (1–479)	33 (2–374)	0.614
Peripheral smear—Blast (%)	55.00 (8–100)	0.5 (0.0–90.0)	0.000 *
Bone marrow—Blast (%)	80.0 (20–100.0)	8.0 (1.0–95.0)	0.000 *

* indicates statistically significant

**Table 4 medicina-59-00498-t004:** Causes of mortality.

Cause of Death	Number of Patients	%
Septic shock	8	33.3
Opportunistic lung infection	3	12.5
COVID-19 infection	1	4.2
Myocardial infarction	1	4.2
Rhinocerebral mucormycosis	1	4.2
GVHD (Graft-versus-Host Disease)	2	8.4

**Table 5 medicina-59-00498-t005:** Cytogenetic risk assessment and immune phenotyping.

Cytogenetic Risk Assessment	*n* (%)
Intermediate-risk group	15 (62.5%)
Intermediate-risk group (F)	6 (25%)
Intermediate-risk group (M)	9 (37.5%)
Favorable-risk group	9 (37.5%)
Favorable-risk group (F)	2 (8.3%)
Favorable-risk group (M)	7 (29.2%)
Immune phenotyping	
CD33(+)	24 (100%)
CD34(+)	11 (45.8%)
CD13(+)	16 (66.6%)
CD117(+)	19 (79.17%)
MPO(+)	17 (70.8%)

**Table 6 medicina-59-00498-t006:** Distribution of adverse events associated with GO.

	Distribution of Adverse Events
Nausea/Vomiting	19 (79.17%)
Headache	15 (62.50%)
Elevated LFT	9 (37.50%)
Febrile Neutropenia	6 (25.00%)
Bleeding	6 (25.00%)
Loss of appetite	4 (16.67%)
Sepsis	4 (16.67%)
Allergy	3 (12.50%)
Tumor Lysis Syndrome	3 (12.50%)
Opportunistic Infection	3 (12.50%)
QT Prolongation	3 (12.50%)
Hepatic Veno-occlusive Disease	2 (8.33%)
Pruritus	2 (8.33%)
Erythema	2 (8.33%)
Cytokine Release Syndrome	1 (4.17%)
Urinary Tract Infection	1 (4.17%)

## Data Availability

The datasets used and/or analyzed during the current study are available from the corresponding author on reasonable request.

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
