# Peer review of "The Use of Gemtuzumab Ozogamicin as Salvage Therapy in Patients with Acute Myeloid Leukemia: A Monocentric Real-World Experience"

_medicina, 2023, doi:10.3390/medicina59030498_

Round 1

Reviewer 1 Report

Dear authors, Materials and Methods should include the cytogenetic risk assessment modality/mutational analysis of the patients, as well as those immunophenotyping markers used to determine the blast rate and prognosis. Will you please provide the results of cytogenetic/mutational analysis?

Author Response

Dear Editors and Reviewers,

I was criticized for not describing the methods adequately and the results not being presented clearly. I realized my mistake and corrected the points you mentioned in the article. Your reviews really helped me a lot.

There was a very strong earthquake in our region on Monday, February 6, 2023, at 04:17. I apologize if I gave an incomplete response to your criticism.

Kind regards

Reviewer 2 Report

Gemtuzumab ozogamicin(GO) for relapsed/refractory acute myeloid leukemia was rediscovered after dose modification. Currently, there have been clinical trials that combine GO with other drugs. It could be meaningful if real-world data were sophisticatedly described for comparison with future trials.

 However, there are some problems in this manuscript. It doesn’t seem to have novelty. There has been suggested AML risk stratification update to ELN2022, and FLT3-ITD mutation status is essential to evaluated outcomes on patients with AML. This manuscript doesn't find meaning in the current risk group.

Moreover, in the case of analyzing Non-relapsed mortality, you need to consider competing risk with the relapse. In addition to the lack of these detailed statistical presentations, there seem to be some critical errors in the results on OS and DFS.

Gemtuzumab ozogamicin(GO) for relapsed/refractory acute myeloid leukemia was rediscovered after dose modification. Currently, there have been clinical trials that combine GO with other drugs. It could be meaningful if real-world data were sophisticatedly described for comparison with future trials.

 However, there are some problems in this manuscript. It doesn’t seem to have novelty. There has been suggested AML risk stratification update to ELN2022, and FLT3-ITD mutation status is essential to evaluated outcomes on patients with AML. This manuscript doesn't find meaning in the current risk group.

Moreover, in the case of analyzing Non-relapsed mortality, you need to consider competing risk with the relapse. In addition to lack of these detailed statistical presentations, there seem to be some critical errors in the results on OS and DFS.

 point by point ------------------------------------------------------------------

2. Materials and Methods

(Line 64) diagnosed according to the WHO 2008 Acute Leukemia Diagnostic Criteria in the Hematology Service of the Department of Internal Medicine, Faculty of Medicine, Çukurova University between 2015 and 2021 were analyzed

3. Resultã„´

(Line 93)

The dose of GO was 3 mg/m2, which was administered on days 1, 4, and 7. 93 Gemtuzumab was used in combination with cytarabine (100-200 mg/m2/day) in one patient. Other concomitant medicines used were pheniramine hydrogen maleate, dexamethasone, and paracetamol.

Question 1

In 2000, GO was approved with 9 mg/m2 repeated in 14 days. In contrast, in 2017, the infusion dose was changed to 3 mg/m2 on Days 1, 4, and 7. However, the patient was diagnosed in 2015. I wonder if all patients were infused with the same dose.

3. Result

(Line 91) The number of patients with only relapsed AML, only refractory AML and r/r AML was 2 (8.3%), 9 (37.5%), and 13 (54.2%), respectively.

Question 2

Thirteen patients with r/r AML definition are unclear and difficult to understand. Does it mean they relapsed after transplantation and were refractory to salvage chemotherapy? If so, a more detailed clinical course for each patient is needed.

3. Results

(Line 120) The median overall survival time of the patients was 44 months (37.8 - 50.2 months). The two-year survival rate was 95%, while the 5-year survival rate was 33%. Disease-free survival was 22 months (0–48.6 months)

Question 3

In the Trial of MyloFrance1, previously published, CR rated were 26%  and DFS was 11.6 months, which is different from the manuscript. It is weird, considering real-world data usually show worse outcomes than prospective trials.

Despite the minimum DFS being zero months, the minimum overall survival was 37.8 months.  Survival analysis based on the time of diagnosis rather than the time of the first relapse may mislead the reader.

3. Result

(Line 115) Cytogenetic analysis showed that 24 patients had a normal karyotype (46 XX or 46 XY). Overall, 15 (62.5%) were in the intermediate-risk cytogenetic group and 9 (37.5%) were in the poor-risk cytogenetic group. Six (75%) female patients had intermediate-risk 117 while two (25%) had poor-risk; and nine (56.25%) male patients had intermediate-risk and 118 seven (43.75%) had poor-risk.

Question 4

AML risk stratification has been presented up to ELN2022. Is it possible to describe additional molecular alterations such as mutations or at least FLT3-ITD status among patients?

4. Discussion

(Line 145) GO has an acceptable toxicity profile in patients with r/r AML which resulted in increased use particularly in frail patients who already received intensive therapy. Acute infusion-related toxicities associated with GO are usually transient and respond to standard treatment of care.

Question 5

To argue that GO toxicity is tolerable, it would be better to present nonrelapse mortality. An analysis considering the competing risk for relapse should be given. Can you show OS, DFS, Non-relapse mortality, and cumulative incidence of relapse from the time of using GO rather than from the time of diagnosis?

Minor

Table2

Respiratory distress: 8 (333%)

è 33.3%

Creatinine (mg/dL) 18.5 (6 – 49.0)

è It sounds weird that most patients had kidney injuries at the initiating treatment.

(Line159) The median follow-up was 44.3 months (13-144). 8 (33.3%) patients were female and 16 (66.7%) were male. The mean age of the patients at diagnosis was 47.63±16.93 (18 - 74) years, with 50.75±15.68 162 years in female patients and 46.06±18.33 years in male patients.

è It seems that the contents of the result are repeated. There is no need to write again in the discussion.

Author Response

Dear Editors and Reviewers,

Care was taken to provide sufficient background and include all relevant references in the introduction. All cited references were checked for relevance to the research. The deficiencies in the research design and methods. The results were tried to be presented clearly and importance was given to supporting the results with results.

Revision has been made, primarily because you indicated that the English language and style should be extensively edited.

In the introduction, the sentences from source 4 to source 5 and source number 5 have been changed in line with your suggestions.

As of line 53, the indications of GO in ELN 2022 have been added in line with your suggestions, indicating the source.

Major changes were made in the material and method part. Added methods from line 72 to line 88.

From line 151 to line 158, cytogenetic and molecular genetic mutations were revised according to your suggestions.

Respiratory distress and high creatinine prevalence were changed according to your recommendations.

You stated that there is not enough information for the introduction and therefore it needs to be improved. Added information about AML risk classification update to ELN2022 introduction section. In addition, information on which patients should be given GO treatment according to the ELN 2022 AML guideline was added. The importance of molecular mutations such as FLT3-ITD was also highlighted. In the conclusion part, frequent risk-determining mutation analyzes in our patients were added.

Thanks to your criticism, we noticed the lack of diagnostic criteria for acute leukemias and the dates of the study. GO was administered as 3 mg/m2 on days 1, 4 and 7 in all patients as standard.

Information on patients with r/r AML definition was explained more clearly.

Question 3.

In the Trial of MyloFrance1, previously published, CR rated 26% and DFS was 11.6 months, which is different from the manuscipt. It is weird, considering real-world data usually show worse outcomes than the prospective trials.

Despite the minimum DFS being zero months, the minimum overall survival was 37.8 months. Survival analysis based on the time of diagnosis rather than the time of the first relapse may mislead the reader.

Response 3.

Compared to many studies in literature, the sample size in this study is very low. There may be slight deviations in the calculation of survival rate. It would be more accurate to give only the 5-year survival rate. However, a meta- analysis of 5 randomized trials showed that the addition of GO to chemotherapy improved the estimated 5- year survival from 50% to 75%. (Hills RK, Castaigne S, Appelbaum FR, et al. Addition of gemtuzumab ozogamicin to induction chemotherapy in adult patients with acute myeloid leukaemia: a meta- analysis of individual patient data from randomised controlled trials. Lancet Oncol. 2014;15:986-996.)

For this reason, the 5-year survival rate added to line 139. In the discussion section, this was stated in line 205.

Low dose GO term was used instead of GO between line 149 and line 151. Source 26 has been added to the end of the sentence. Added reference number 27 for the sentence from line 151 to line 152. The location of all references has been checked.

Knowledge of changes involving molecular mutations among patients was noted in the article.

Question 5.

To argue that GO toxicity is tolerable, it would be better to present nonrelapse mortality. An analysis considering the competing risk for relapse should be given. Can you show OS, DFS, Non—relapse mortality, and cumulative incidence of relapse from the time of using GO rather than from the time of the diagnosis?

Response 5.

References are included in the first part of the discussion for information cited from other studies.

“Low dose GO has an acceptable toxicity profile in patients with r/r AML which resulted in increased use particularly in frail patients who already received intensive therapy. (Domingo, M. P., Polo, S. V., Díaz, A. G., Jordà, R. C., Santasusana, J. R., & Coll, C. F. (2021). Low doses of gemtuzumab ozogamicin in adults diagnosed with acute myeloid leukaemia. Medicina Clínica (English Edition), 157(7), 325-328.) Acute infusion-related toxicities associated with GO are usually transient and respond to standard treatment of care. (Appelbaum, F. R., & Bernstein, I. D. (2017). Gemtuzumab ozogamicin for acute myeloid leukemia. Blood, The Journal of the American Society of Hematology, 130(22), 2373-2376.)”

In patients with respiratory distress and elevated creatinine, there was an error in writing. For example, creatinine is written as 18. In fact, the BUN 18 value was written as creatinine at the time of writing the article. Thanks to being very careful like you, this error has been fixed. I really, really found your criticism extremely important. I believe in correcting my mistakes and with my best regards.

There was a very strong earthquake in our region on Monday, February 6, 2023, at 04:17. I apologize if I gave an incomplete response to your criticism.

Round 2

Reviewer 2 Report

The commented reference[1] showed that the median overall survival(OS) was 22.5 months,  and 5-year OS  was 34%.

Did you check that overall survival was calculated at the initiation of salvage treatment, not from the AML diagnosis?

I doubt the core outcomes of the manuscript, so hard to agree with publishing.  

[1] Hills RK et al. Addition of gemtuzumab ozogamicin to induction chemotherapy in adult patients with acute myeloid leukaemia: a meta-analysis of individual patient data from randomised controlled trials. Lancet Oncol. 2014 Aug;15(9):986-96. doi: 10.1016/S1470-2045(14)70281-5. Epub 2014 Jul 6. PMID: 25008258; PMCID: PMC4137593.

Author Response

16.02.2023

Dear Reviewer,

Thank you for your comment. Your comments are very valuable for us.

You can see our response to your comment below:

Reviewer Comment

The commented reference (1) showed that the median overall survival (OS) was 22.5 months, and 5-year OS was 34%.

Did you check that overall survival was calculated at the initiation of salvage treatment, not from the AML diagnosis?

Response to the reviewer

The study you cited (Hills RK, et al.  Addition of gemtuzumab ozogamicin to induction chemotherapy in adult patients with acute myeloid leukaemia: a meta-analysis of individual patient data from randomised controlled trials. Lancet Oncol. 2014 Aug;15(9):986-96. doi: 10.1016/S1470-2045(14)70281-5. Epub 2014 Jul 6. PMID: 25008258; PMCID: PMC4137593.) is a meta-analysis with 3,325 patients. The meta-analysis was done on controlled clinical trials with newly-diagnosed AML or MDS with large number of patients in multiple centers over 15 years of age. On the other hand, this study included 24 AML patients who had already been treated with GO in a single-center, with retrospective data- as a real world experience. We have analyzed the data again and our results are in line with the references we have already inserted in the manuscript.

We have read the manuscript again and you can see the updates that we have done in the manuscript with track changes.

Kindest regards,
